# Response Analysis of the Three-Degree-of-Freedom Vibroimpact System with an Uncertain Parameter

**DOI:** 10.3390/e25091365

**Published:** 2023-09-21

**Authors:** Guidong Yang, Zichen Deng, Lin Du, Zicheng Lin

**Affiliations:** 1Department of Engineering Mechanics, Northwestern Polytechnical University, Xi’an 710129, China; dweifan@nwpu.edu.cn; 2School of Mathematics and Statistics, Xidian University, Xi’an 710071, China; 2022103676@ruc.edu.cn; 3School of Mathematics and Statistics, Northwestern Polytechnical University, Xi’an 710129, China; lindu@nwpu.edu.cn

**Keywords:** vibroimpact system, bifurcation, Chebyshev polynomial, uncertain parameter

## Abstract

The inherent non-smoothness of the vibroimpact system leads to complex behaviors and a strong sensitivity to parameter changes. Unfortunately, uncertainties and errors in system parameters are inevitable in mechanical engineering. Therefore, investigations of dynamical behaviors for vibroimpact systems with stochastic parameters are highly essential. The present study aims to analyze the dynamical characteristics of the three-degree-of-freedom vibroimpact system with an uncertain parameter by means of the Chebyshev polynomial approximation method. Specifically, the vibroimpact system model considered is one with unilateral constraint. Firstly, the three-degree-of-freedom vibroimpact system with an uncertain parameter is transformed into an equivalent deterministic form using the Chebyshev orthogonal approximation. Then, the ensemble means responses of the stochastic vibroimpact system are derived. Numerical simulations are performed to verify the effectiveness of the approximation method. Furthermore, the periodic and chaos motions under different system parameters are investigated, and the bifurcations of the vibroimpact system are analyzed with the Poincaré map. The results demonstrate that both the restitution coefficient and the random factor can induce the appearance of the periodic bifurcation. It is worth noting that the bifurcations fundamentally differ between the stochastic and deterministic systems. The former has a bifurcation interval, while the latter occurs at a critical point.

## 1. Introduction

Collisions and impacts are common physical phenomena in complex engineering structures and systems, which can significantly affect their stability, reliability, and even cause irreversible damage to mechanical equipment. Such systems, referred to as vibroimpact systems, are typically described using differential equations with constraints due to the non-smooth nature of collisions and impacts [1,2,3]. This inherent non-smoothness introduces strong nonlinearity into the system dynamics, leading to complex behaviors and a high sensitivity to parameter changes [4,5]. Moreover, engineering structures are inevitably influenced by random factors such as air temperature variations, humidity changes, and ground vibrations [6,7,8,9]. These factors can introduce randomization characteristics to the system parameters, making the analysis of dynamic properties even more challenging. Thus, studying the stochastic dynamics of vibroimpact systems is not only interesting but also crucial, albeit facing significant challenges.

Due to the non-smoothness and nonlinearity of vibroimpact systems, closed-form solutions are generally unavailable [10]. Numerical simulations play a crucial role in understanding and analyzing the dynamics of these systems [11,12,13]. By employing a mapping that relates conditions at subsequent impacts, Shaw and Holmes [14,15,16] discussed the dynamical behaviors of harmonically excited impact systems, including periodic motions, harmonic and subharmonic motions, chaotic motions, and global bifurcations. Mason [17] analyzed codimension-one, -two and -three bifurcations in a periodically forced impact oscillator with two discontinuity surfaces using discontinuity-geometry methodology. Li [18] investigated the effects of various system parameters on the stochastic P-bifurcation phenomenon in a vibroimpact system using path integration and the generalized cell mapping method. Liu [19] studied the crises in the Duffing vibroimpact oscillator with non-viscously damping via the composite cell coordinate system method. Qian [20] utilized the radial basis function neural networks method to analyze typical randomly excited vibroimpact systems. Ding [21] established a six-dimensional Poincaré map to explore the double Neimark–Sacker bifurcation, torus T2 of a three-degree-of-freedom vibro-impact system. Hopf bifurcation is a common bifurcation phenomenon, mainly occurring in dynamical systems, particularly in systems that describe oscillatory behaviors [22]. At a Hopf bifurcation point, the behavior of the system changes from a stable, unchanging state to a periodic or oscillatory state. The Hopf bifurcation phenomenon of vibroimpact systems has also been reported [23,24]. As research on vibroimpact systems deepens, unique dynamic phenomena have been discovered, including grazing bifurcations [25,26,27], stick–slip motion [28,29], chattering [30,31], and corner point bifurcation [32,33].

As the study of vibroimpact systems progresses, it becomes evident that these systems exhibit strong parameter sensitivity dependencies [34,35,36,37]. This implies that even small variations or adjustments in the system parameters can have a significant impact on the system’s dynamic behavior. Unfortunately, uncertainties and errors are inevitable in various stages such as modeling, measurement, and manufacturing. Even minor deviations can result in fundamental alterations in the dynamic characteristics of the system. Consequently, studying vibroimpact systems with stochastic parameters becomes highly essential. Sampaio [38] investigated the transient dynamics of an uncertain vibroimpact system, and the robustness of the predictions was analyzed with respect to model uncertainties. Lima [39] analyzed the maximal energy stored in a barrier due to the impacts of a pendulum fitted within a vibroimpact electromechanical system considering the existence of epistemic uncertainties in the system parameters. Feng [40] studied the period-doubling phenomenon of a single-degree-of-freedom vibroimpact system with an uncertain parameter. However, actual models established through mechanical relationships are usually vibroimpact systems with multiple degrees of freedom. The high complexity and parameter uncertainty of vibroimpact systems pose great difficulties in studying the dynamic characteristics of the systems. How to theoretically transform uncertain parameters while maintaining the main characteristics of system dynamics remains a crucial technical challenge. The Chebyshev polynomial approximation method has been proven to be an effective method for handling smooth systems with uncertain parameters [41,42]. As for non-smooth systems, especially multiple-degrees-of-freedom vibroimpact systems, there are few studies focusing on the dynamic characteristics of these systems via this approximation method. Because the three-degree-of-freedom vibroimpact systems offer a good balance between computational cost, model accuracy, and theoretical insight, we aim to analyze the dynamic response of a three-degree-of-freedom vibroimpact system with an uncertain parameter using polynomial approximation methods. The results indicate that our proposed method is very effective. The influences of different parameters on the bifurcation behavior of the system using this method have been discussed in detail. An interesting phenomenon is discovered through the research: the overall periodic response tends to remain relatively stable in the presence of minor stochastic influences. However, at critical junctures where period-doubling bifurcations arise, even slight random fluctuations can exert a substantial impact on system behavior.

The paper is organized as follows: Section 2 presents the model of a three-degree-of-freedom vibroimpact system with an uncertain parameter; in Section 3, the Chebyshev polynomial approximation is utilized to transform the vibroimpact system with an uncertain parameter into an equivalent deterministic system; Section 4 provides numerical results to demonstrate the effectiveness of our method and analyzes the influence of system parameters on the dynamic behavior of the system; finally, the paper concludes with key findings.

## 2. Model of the Three-Degree-of-Freedom Vibroimpact System with an Uncertain Parameter

Vibroimpact systems often involve multiple impacts with rigid barriers, and the model represented by differential equations with constraints is referred to as the classical impact model. A schematic of the three-degree-of-freedom vibroimpact system with unilateral rigid constrain [43] is shown in Figure 1. The stiffness coefficients of the three springs are $K_{1},K_{2},K_{3}$. Three masses $M_{1},M_{2},M_{3}$ are attached to the wall using these three springs. Simultaneously, three linear dampers with damping coefficients, $C_{1},C_{2},C_{3}$, are connected to the masses at the same time. Above every mass, there is a harmonic excitation force, $P_{i}\sin(\omega t+\tau )\{i=1,2,3\}$. When the first mass, $M_{1}$, moves the distance of B, it will impact with the rigid barrier, A. The energy loss during the impact is determined using the restitution coefficient.

The dimensionless form of the system motion differential equation can be written as
(1)$$M\ddot{X}+2C\dot{X}+KX=P\sin(\omega t+\tau ),\;(x_{1}<b)$$
(2)$${\dot{x}}_{1A+}=-r{\dot{x}}_{1A-},\;(x_{1}=b)$$
where, $\vec{X}={\left(x_{1},x_{2},x_{3}\right)}^{T}$, $\vec{P}={\left(p_{1},p_{2},p_{3}\right)}^{T}$, $\vec{M}=\left[\begin{array}{ccc} m_{1} & 0 & 0\\ 0 & m_{2} & 0\\ 0 & 0 & m_{3} \end{array}\right]$, $\vec{K}=\left[\begin{array}{ccc} k_{1} & -k_{1} & 0\\ -k_{1} & k_{1}+k_{2} & -k_{2}\\ 0 & -k_{2} & k_{2}+k_{3} \end{array}\right]$, $\vec{C}=\left[\begin{array}{ccc} \zeta_{1} & -\zeta_{1} & 0\\ -\zeta_{1} & \zeta_{1}+\zeta_{2} & -\zeta_{2}\\ 0 & -\zeta_{2} & \zeta_{2}+\zeta_{3} \end{array}\right]$.

During dimensionless form transformation, some assumptions have been made such as $M_{1}\neq 0$, $K_{1}\neq 0$ and $P_{0}=\sqrt{P_{1}{}^{2}+P_{2}{}^{2}+P_{3}{}^{2}}$, and the dimensionless quantities are $m_{i}=\frac{M_{i}}{M_{1}}$, $k_{i}=\frac{K_{i}}{K_{1}}$, $p_{i}=\frac{P_{i}}{P_{1}}$, $\zeta_{i}=\frac{C_{i}}{2\sqrt{K_{1}M_{1}}}$, $\omega =\Omega \sqrt{\frac{M_{1}}{K_{1}}}$, $t=T\sqrt{\frac{K_{1}}{M_{1}}}$, $b=\frac{BK_{1}}{P_{0}}$, $x_{i}=\frac{X_{i}K_{1}}{P_{0}}$, and i=1,2,3.

Note that the linear damper $C_{i}$ is considered to be the random parameter of the system. Thus, $\zeta_{i}$ can be reduced to the form of $\zeta_{i}={\bar{\zeta }}_{i}+\delta_{i}U$, in which ${\bar{\zeta }}_{i}$ and $\frac{\delta_{i}}{2}$ are the mean value and variance of $m_{i}$, respectively, and U is the random variable whose probability density function is defined on [−1,1] following arch distribution. $x_{i}$ represents the dimensionless displacement of each mass block relative to its initial position. When the mass $M_{1}$ arrives at the position of the rigid barrier, A, its velocity will change abruptly, and r is Newton’s restitution coefficient, which is used to characterize the magnitude of energy loss during the impact instant. ${\dot{x}}_{1A+}$ and ${\dot{x}}_{1A-}$ represent the velocity before and after the impact, respectively.

## 3. The Approximation of the Vibroimpact System with an Uncertain Parameter

### 3.1. Chebyshev Polynomial Approximation

Orthogonal polynomial approximation is an effective method to solve complex nonlinear equations. However, the selection of a polynomial basis must be consistent with the probability density function satisfied by the random parameters in the equation. Though the normal distribution corresponding to Hermite polynomials will provide us with a useful procedure to deal with the case that system parameters obeys Gaussian distribution, the value range of normal distribution random variables is from negative infinity to positive infinity, which is contrary to the actual situation. Therefore, the random parameter is considered the random variable subject to arch distribution, which not only meets the implementation situation, but also will not cause the phenomenon of system instability. The probability density function of random variables satisfying the arch distribution is as follows:(3)$$p(u)=\left\{\begin{array}{ll} \frac{2}{\pi }\sqrt{1-u^{2}} & \left|u\right|\leq 1,\\ 0 & \left|u\right|>1. \end{array}\right.$$

The curve of p(u) is shown in Figure 2. It is easy to see from the figure that the random parameter is bounded, which is more reasonable in mechanical engineering.

The corresponding polynomial of arch distribution is the Chebyshev polynomial of the second kind, and its general expression is
(4)$$U_{n}(u)=\sum_{i=0}^{\frac{n}{2}}{\left(-1\right)}^{i}\frac{(n-i)!}{i!(n-2i)!}{\left(2u\right)}^{n-2i}.$$

Applying Equation (4), we have
(5)$$\begin{array}{c} U_{0}(u)=1\\ U_{1}(u)=2u\\ U_{2}(u)=4u^{2}-1\\ U_{3}(u)=8u^{3}-4u\\ U_{4}(u)=16u^{4}-12u^{2}+1\\ \ldots \end{array}$$

In addition, the second kind of Chebyshev polynomials also have a recursive formula:(6)$$uU_{i}(u)=\frac{1}{2}[U_{i-1}(u)+U_{i+1}(u)].$$

The second type of Chebyshev polynomials has orthogonality, whose weighted orthogonality can be expressed as
(7)$$\int_{-1}^{1}\frac{2}{\pi }\sqrt{1-u^{2}}U_{i}(u)U_{j}(u)du=\left\{\begin{array}{l} 1\;(i=j),\\ 0\;(i\neq j). \end{array}\right.$$

Equation (7) represents the weighted orthogonal relation of polynomials. Although u is a random variable between −1 and +1, since the weight function is the same as the probability density function of the random variable u, the left side of Equation (7) can be considered to be the expectation of the random function $U_{i}(u)U_{j}(u)$. According to the nature of function of random variable, randomness does not affect the orthogonality of polynomials. Moreover, due to the orthogonality of the second kind of Chebyshev polynomials, any measurable function, f(u), of random variable u can be expanded into the following series:(8)$$f(u)=\sum_{i=0}^{\infty }c_{i}U_{i}(u),$$
$c_{i}=\frac{\int_{-\infty }^{\infty }p(u)f(u)U_{i}(u)du}{\int_{-\infty }^{\infty }p(u)U_{i}(u)U_{i}(u)du}=\int_{-1}^{1}p(u)f(u)U_{i}(u)du$.

This expansion method, based on the generalized Fourier series, is called the orthogonal decomposition of random function, f(u), and this approximation method is the best for mean square approximation.

### 3.2. Equivalent Deterministic System

In this section, Equation (1) will be approximated using an orthogonal polynomial. Under unconstrained conditions, the response of the system can be expressed as a function of time, t, and random variable u, and can be expanded into the following series form:(9)$$\vec{X}(t,u)=\sum_{t=0}^{\infty }{\vec{X}}_{i}(t){\vec{U}}_{i}(u).$$

When i only takes a finite number N, Equation (9) can be approximately expressed as
(10)$$\vec{X}(t,u)\approx \sum_{t=0}^{N}{\vec{X}}_{i}(t){\vec{U}}_{i}(u).$$
where, ${\vec{U}}_{i}(u)$ represents the three-dimensional vector composed of the *i*-th Chebyshev polynomial. We also have $X_{i}(t)=\int_{-1}^{1}p(u)X(t,u)U_{i}(u)du$, in which $X_{i}(t)={\left(X_{1i},X_{2i},X_{3i}\right)}^{T}$.

When the impact factor is not taken into account, substituting Equation (10) into Equation (1) yields the following result:(11)$$\vec{M}\frac{d^{2}\sum_{i=0}^{N}{\vec{X}}_{i}(t){\vec{U}}_{i}(u)}{dt^{2}}+2\vec{C}\frac{d\sum_{i=0}^{N}{\vec{X}}_{i}(t){\vec{U}}_{i}(u)}{dt}+\vec{K}\sum_{i=0}^{N}{\vec{X}}_{i}(t){\vec{U}}_{i}(u)=\vec{P}\sin(\omega t+\tau ).$$

By dividing the vector form of Equation (11) into scalar form, three equations can be obtained:(12)$$\begin{array}{c} \left(m_{1}\frac{d^{2}}{dt^{2}}+k_{1})\sum_{i=0}^{N}X_{1i}(t)U_{i}(u)-k_{1}\sum_{i=0}^{N}X_{2i}(t)U_{i}(u\right)\\ -2\bar{\zeta_{1}}\frac{d}{dt}\sum_{i=0}^{N}X_{1i}(t)U_{i}(u)+2\delta_{1}u\frac{d}{dt}\sum_{i=0}^{N}X_{1i}(t)U_{i}(u)\\ -2\bar{\zeta_{1}}\frac{d}{dt}\sum_{i=0}^{N}X_{2i}(t)U_{i}(u)-2\delta_{1}u\frac{d}{dt}\sum_{i=0}^{N}X_{2i}(t)U_{i}(u)=p_{1}\sin(\omega t+\tau ) \end{array}$$



(13)
$$\begin{array}{c} \left(m_{2}\frac{d^{2}}{dt^{2}}+k_{1}+k_{2}\right)\sum_{i=0}^{N}X_{2i}\left(t\right)U_{i}(u)-k_{1}\sum_{i=0}^{N}X_{1i}\left(t\right)U_{i}(u)-k_{2}\sum_{i=0}^{N}X_{3i}\left(t\right)U_{i}(u)\\ +2\left(\bar{\zeta_{1}}+\bar{\zeta_{2}}\right)\frac{d}{dt}\sum_{i=0}^{N}X_{2i}\left(t\right)U_{i}(u)-2\bar{\zeta_{1}}\frac{d}{dt}\sum_{i=0}^{N}X_{1i}\left(t\right)U_{i}(u)\\ -2\bar{\zeta_{2}}\frac{d}{dt}\sum_{i=0}^{N}X_{3i}\left(t\right)U_{i}(u)+2\left(\delta_{1}+\delta_{2}\right)u\frac{d}{dt}\sum_{i=0}^{N}X_{2i}\left(t\right)U_{i}(u)\\ -2\delta_{1}u\frac{d}{dt}\sum_{i=0}^{N}X_{1i}\left(t\right)U_{i}(u)-2\delta_{2}u\frac{d}{dt}\sum_{i=0}^{N}X_{3i}\left(t\right)U_{i}(u)=p_{2}\sin\left(\omega t+\tau \right) \end{array}$$





(14)
$$\begin{array}{c} \left(m_{3}\frac{d^{2}}{dt^{2}}+k_{2}+k_{3})\sum_{i=0}^{N}X_{3i}(t)U_{i}(u)-k_{2}\sum_{i=0}^{N}X_{2i}(t)U_{i}(u)-2\bar{\zeta_{2}}\frac{d}{dt}\sum_{i=0}^{N}X_{2i}(t)U_{i}(u\right)\\ +2(\bar{\zeta_{2}}+\bar{\zeta_{3}})\frac{d}{dt}\sum_{i=0}^{N}X_{3i}(t)U_{i}(u)+2(\delta_{2}+\delta_{3})u\frac{d}{dt}\sum_{i=0}^{N}X_{3i}(t)U_{i}(u)\\ -2\delta_{2}u\frac{d}{dt}\sum_{i=0}^{N}X_{2i}(t)U_{i}(u)=p_{3}\sin(\omega t+\tau ) \end{array}$$



The recursive relation equation, Equation (6), of a Chebyshev polynomial can be obtained as
(15)$$\begin{array}{c} u\sum_{i=0}^{N}X_{i}(t)U_{i}(u)=\frac{1}{2}\sum_{i=0}^{N}X_{i}(t)[U_{i-1}(u)+U_{i+1}(u)]\\ =\frac{1}{2}\sum_{i=0}^{N}U_{i}(u)[X_{i-1}(t)+X_{i+1}(t)]. \end{array}$$

Due to the approximation of Equation (10), $X_{-1}$ and $X_{N+1}$ are set to zero. By substituting Equation (15) into each of the expressions in Equations (12)–(14), and then multiplying both ends of each of the resulting equations by $U_{i}(u)(i=0,\ldots ,N)$, and finally taking the expectation with regard to the random variable, u, the following result is obtained:
(16)$$\left\{\begin{array}{c} \left(m_{1}\frac{d^{2}}{dt^{2}}+k_{1}\right)X_{10}\left(t\right)-k_{1}X_{20}\left(t\right)+2\bar{\zeta_{1}}\frac{d}{dt}X_{10}\left(t\right)+\delta_{2}\frac{d}{dt}X_{11}\left(t\right)=p_{1}\sin\left(\omega t+\tau \right)\\ \left(m_{2}\frac{d^{2}}{dt^{2}}+k_{1}+k_{2}\right)X_{20}\left(t\right)-k_{1}X_{10}\left(t\right)-k_{3}X_{30}\left(t\right)+2\left(\bar{\zeta_{1}}+\bar{\zeta_{2}}\right)\frac{d}{dt}X_{20}\left(t\right)\\ -2\bar{\zeta_{1}}\frac{d}{dt}X_{10}\left(t\right)-2\bar{\zeta_{2}}\frac{d}{dt}X_{30}\left(t\right)+\left(\delta_{1}+\delta_{2}\right)\frac{d}{dt}X_{21}\left(t\right)\\ -\delta_{1}\frac{d}{dt}X_{11}\left(t\right)-\delta_{2}\frac{d}{dt}X_{31}\left(t\right)=p_{2}\sin\left(\omega t+\tau \right)\\ \left(m_{3}\frac{d^{2}}{dt^{2}}+k_{1}+k_{2}\right)X_{30}\left(t\right)-k_{2}X_{20}\left(t\right)+2\left(\bar{\zeta_{3}}+\bar{\zeta_{2}}\right)\frac{d}{dt}X_{30}\left(t\right)-2\bar{\zeta_{2}}\frac{d}{dt}X_{20}\left(t\right)\\ +\left(\delta_{3}+\delta_{2}\right)\frac{d}{dt}X_{31}\left(t\right)-\delta_{2}\frac{d}{dt}X_{21}\left(t\right)=p_{3}\sin\left(\omega t+\tau \right)\\ \left(m_{1}\frac{d^{2}}{dt^{2}}+k_{1}\right)X_{1i}\left(t\right)U_{i}\left(u\right)-k_{1}X_{2i}\left(t\right)+2\bar{\zeta_{1}}\frac{d}{dt}X_{1i}\left(t\right)-2\bar{\zeta_{1}}\frac{d}{dt}X_{2i}\left(t\right)\\ +\delta_{1}\frac{d}{dt}\left(X_{1i-1}\left(t\right)+X_{1i+1}\left(t\right)\right)-\delta_{1}\frac{d}{dt}\left(X_{2i-1}\left(t\right)+X_{2i+1}\left(t\right)\right)=0\\ \left(m_{2}\frac{d^{2}}{dt^{2}}+k_{1}+k_{2}\right)X_{2i}\left(t\right)-k_{1}X_{1i}\left(t\right)-k_{2}X_{3i}\left(t\right)\\ +2\left(\bar{\zeta_{1}}+\bar{\zeta_{2}}\right)\frac{d}{dt}X_{2i}\left(t\right)U_{i}\left(u\right)-2\bar{\zeta_{1}}\frac{d}{dt}X_{1i}\left(t\right)-2\bar{\zeta_{2}}\frac{d}{dt}X_{3i}\left(t\right)U_{i}\left(u\right)\\ +\left(\delta_{1}+\delta_{2}\right)\frac{d}{dt}\left(X_{2i-1}\left(t\right)+X_{2i+1}\left(t\right)\right)-\delta_{1}\frac{d}{dt}\left(X_{1i-1}\left(t\right)+X_{1i+1}\left(t\right)\right)\\ -\delta_{2}\frac{d}{dt}\left(X_{3i-1}\left(t\right)+X_{3i+1}\left(t\right)\right)=0\\ \left(m_{3}\frac{d^{2}}{dt^{2}}+k_{2}+k_{3}\right)X_{3i}\left(t\right)U_{i}\left(u\right)-k_{2}X_{2i}\left(t\right)-2\bar{\zeta_{2}}\frac{d}{dt}X_{2i}\left(t\right)\\ +2\left(\bar{\zeta_{3}}+\bar{\zeta_{2}}\right)\frac{d}{dt}X_{3i}\left(t\right)+\left(\delta_{3}+\delta_{2}\right)\frac{d}{dt}\left(X_{3i-1}\left(t\right)+X_{3i+1}\left(t\right)\right)U_{i}\left(u\right)\\ -\delta_{2}\frac{d}{dt}\left(X_{2i-1}\left(t\right)+X_{2i+1}\left(t\right)\right)=0\;(i=1,...,N) \end{array}\right.$$

Therefore, the unconstrained three-degree-of-freedom unilateral rigid constraint vibroimpact system Equation (1) is reduced to an equivalent deterministic system Equation (16).

According to the approximation of Equation (10), the ensemble mean response of the system can be approximated as
(17)$$E[\vec{X}(t,u)]=E[\sum_{i=0}^{N}{\vec{X}}_{i}(t){\vec{U}}_{i}(u)]=\sum_{i=0}^{N}{\vec{X}}_{i}(t)E[{\vec{U}}_{i}(u)]=X_{0},$$
where $\vec{X_{0}}={\left(x_{10},x_{20},x_{30}\right)}^{T}$.

Note that the random variable u takes a random value of $\left\{u_{i}\right\}$ on the interval [−1,1], and that there is a certain time, t, in the constrained case in which some orbits of the sample system have impacted with the constraint plane, ∑(b) (the virtual constraint plane here), while some have not reached the constraint plane, ∑(b), due to the influence of random factors. Therefore, it is necessary to average the constraint conditions.

According to Equation (17), the average response can be obtained. The average constraint surface is as follows:(18)$$\bar{\sum }=\{(\vec{X},\vec{\dot{X}})\left|\vec{X_{0}}=\vec{b}\right.\}.$$

The average constraint condition is as follows:(19)$$E[\vec{X}(t,u)]=\vec{X_{0}}<\vec{b}.$$

The average jump equation is as follows:(20)$$\vec{{\dot{X}}_{0+}}=-r\vec{{\dot{X}}_{0-}}.$$

Based on the above Chebyshev polynomial approximation, the introduced average constraint conditions, Equation (19), and the average jump, Equation (20), the stochastic three-degree-of-freedom vibroimpact system is transformed into an equivalent deterministic system. When N→∞, the solution $\sum_{i=0}^{N}{\vec{X}}_{i}(t){\vec{U}}_{i}(u)$ approximates $\vec{X}(t,u)$ in the sense of the best square error. The value of N determines the accuracy of the solution. In this paper, N=2 is taken to obtain an equivalent deterministic unilateral constraint system.

## 4. Reponses of the Three-Degree-of-Freedom Vibroimpact System

### 4.1. Period-Doubling Bifurcation

Through the analysis of Equations (1) and (10), three systems can be obtained; one is the deterministic three-degree-of-freedom vibroimpact system in which the amplitude of the random response is δ=0; another is the original stochastic three-degree-of-freedom vibroimpact system; and the last one is the equivalent deterministic system. The corresponding responses of the three systems are the deterministic system response (DSR), stochastic system response (SSR) and equivalent system response (ESR), respectively. The response of the original stochastic system is obtained via random simulation using the acceptance-rejection method (ARM) algorithm and Monte Carlo method, while responses of DSR and ESR are obtained using numerical simulation methods of deterministic systems. The phase orbit diagrams of the responses of the three systems are compared via numerical simulation to study the dynamical behaviors of the stochastic three-degree-of-freedom vibroimpact system. The maximum Lyapunov exponent is a state quantity that can effectively judge the chaotic state of the system. At present, the Lyapunov exponent method for smooth systems has been relatively mature. For vibroimpact systems, due to the discontinuity induced in the Jacobian matrix by impact factors, the calculation method of smooth systems cannot be directly applied. In this paper, a Lyapunov exponent calculation method for non-smooth systems based on discontinuous mapping is used to calculate the Lyapunov exponent.

In order to investigate the effect of the frequency of resonant forces, some parameters of the system are taken as follows: the amplitude of random parameter δ=0.01, the restitution coefficient r=0.7, the impact condition b=0.7, and resonant forces $P={\left(1,0,0\right)}^{T}$, $m_{2}=m_{3}=1.5$, $k_{2}=k_{3}=4$. Since δ is a small quantity, the initial values of both the stochastic system and the deterministic system can be taken as $\vec{X}={\left(1,5,0\right)}^{T}$ and $\vec{\dot{X}}={\left(1,0,0\right)}^{T}$. For the parameter ω=2.25, the system has a steady-state periodic response, and period is 1T for this case, where $T=\frac{2\pi }{\omega }$. This is the so-called 1−1 periodic motion in which n−p represents periodic motion; n represents the number of cycles and p represents the number of impacts with the constraint surface. The time history diagram and phase orbit diagram are shown in Figure 3a and Figure 3b, respectively. The corresponding results for the parameter ω=2.2 are shown in Figure 3c and Figure 3d, respectively. It is easy to see that the 1−1 motion in Figure 3b becomes a stable period 2−2 motion with the frequency change; that is, period-doubling bifurcation has occurred.

To further observe the dynamical behaviors of the three systems, Figure 4 shows the responses when ω=1.65 and ω=1.787. As shown in Figure 4a,b, the system exhibits a stable 1−1 symmetric motion at ω=1.65, which is close to the motion form of simple harmonic motion. When ω=1.787, a 1−1 motion has period-doubling bifurcation and becomes a stable period 2−2 motion, as shown in Figure 4c,d.

In Figure 5, the curve of the maximum Lyapunov exponent is given for different frequencies in Figure 3 and Figure 4. As can be seen from the figure, the maximum Lyapunov exponent at all four positions is stable and less than 0, which means that the system is indeed in a stable state, matching the phase orbit diagrams. However, with the appearance of period-doubling bifurcation, the maximum Lyapunov exponent also becomes larger in response, which means that the degree of chaos of the system increases.

### 4.2. From Period-Doubling Bifurcation to Chaos

Considering the system parameter $P={\left(3,0,0\right)}^{T}$, $m_{2}=m_{3}=2$. The influence on the system responses for different frequency values is investigated. The time history diagram and phase orbit diagram of the system responses are shown in Figure 6. When ω=2.45, the system has a periodic response, and the system motion mode is 1−1, as shown in Figure 6a,b. As the angular frequency of harmonic excitation, P, decreases to 1−1, the system’s motion mode changes to 2−2 after period-doubling bifurcation, as shown in Figure 6c,d. The successive occurrence of such period-doubling bifurcations indicates the appearance of chaos. It can be seen from the maximum Lyapunov exponent shown in Figure 7 that the maximum Lyapunov exponent changes as the parameters change. When the angular frequency of harmonic excitation P decreases to ω=2.21, the maximum Lyapunov exponent becomes a positive number, which means that the system appears chaotic, as shown in Figure 8.

As can be seen from Figure 7 and Figure 8, when other parameters have been determined, as the angular frequency keeps decreasing, the system responses produce an abundant period-doubling bifurcation phenomenon in the process from ω=2.45 to ω=2.21, and finally lead to chaos. The numerical simulation results show that the phase diagrams of the three systems agree well with each other in the path from period-doubling bifurcation to chaos, which reflects the evolution process of the system.

### 4.3. Influence of the Restitution Coefficient

In the stochastic three-degree-of-freedom vibroimpact system, Equations (1) and (2), the restitution coefficient, r, represents the energy loss in the impact process, which will further affect the periodic phenomenon of the system and lead to bifurcation. The parameters are considered as follows: $P={\left(1,0,0\right)}^{T}$, $m_{2}=m_{3}=1.5$, and $k_{2}=k_{3}=4$. The bifurcation diagram with respect to the restitution coefficient, r, for a deterministic three-degree-of-freedom vibroimpact system is shown in Figure 9. Note that r is Newton’s restitution coefficient, and its value range is 0 to 1. The smaller the restitution coefficient, the greater the energy loss of the system. From the figure, it is obvious that the system response experienced an evolution from period-doubling bifurcations to chaos with a restitution coefficient change. The critical point for the first bifurcation is at r=0.295; the second bifurcation occurs at r=0.673; and when r>0.776, the system becomes chaotic.

To clearly show the influence on the system responses for different restitution coefficients, the phase diagrams are presented in Figure 10. Different restitution coefficient values are chosen to see the changes of the system responses. Since smaller restitution coefficient can induce larger energy loss during the impact of the system, the system will become stable more easily for smaller restitution coefficients. As shown in Figure 10a, when r=0.2, the energy left by the system is not large enough to reach the position of the rigid barrier, so a motion state similar to simple harmonic motion will be formed. When r=0.5, the first block has the energy to reach the constraint surface, and the impact phenomenon occurs as shown in Figure 10b. In this case, one impact occurs for one period. When r=0.7, the system exhibits a period-doubling response, as shown in Figure 10c. When r=0.9, the system enters a chaotic state, as shown in Figure 10d.

### 4.4. Influence of the Uncertain Parameter

For the stochastic three-degree-of-freedom vibroimpact system, when the random variable U takes a series of values $\left\{u_{i}\right\}$ in the interval [−1,1], each value of $u_{i}$ corresponds to a certain non-smooth sample system $\prod u_{i}$. If there is a period-doubling bifurcation in the system, a deterministic bifurcation must occur at a critical point. However, a series of sample systems of a stochastic system may not present bifurcation at the same critical point. There must be an interval in which the period-doubling bifurcation may or may not have occurred in the samples of the random system. When the bifurcation parameters pass through this transition interval, almost all samples have bifurcations, and the period-doubling bifurcation of the random system is considered to be completed.

Take system parameters as $P={\left(1,0,0\right)}^{T}$, $m_{2}=m_{3}=1.5$, $k_{2}=k_{3}=4$. With bifurcation parameters changing from ω=2.14 to ω=2.28, the responses of the deterministic three-degree-of-freedom vibroimpact system change from chaos state to reverse period-doubling bifurcation, as depicted in Figure 11a. On the contrary, when bifurcation parameters increase from ω=1.6 to ω=1.85, the evolutionary process leading to chaos from period-doubling bifurcation is shown in Figure 11b. Regardless of which situation, clear bifurcation points can be seen in the deterministic three-degree-of-freedom vibroimpact system in Figure 11.

To investigate the effect of an uncertain parameter, the same system parameters are considered. Time history and phase orbit diagrams of stochastic and deterministic systems are shown in Figure 12. It can be seen from Figure 12a,c that when the parameter ω=2.135, the phase orbit diagrams of the deterministic system and the random system display notable distinctions. The deterministic system has not yet had bifurcation, but the random system has already had bifurcation. Therefore, under the same initial values and parameters, the bifurcation interval of the stochastic system is indeed different from the bifurcation point of the deterministic system. In this bifurcation interval, the phase orbit diagrams of the stochastic vibroimpact system and the deterministic system are quite different. Period-doubling bifurcation has occurred in the former, but not yet in the latter.

## 5. Conclusions

This paper establishes a model of a three-degree-of-freedom unilateral rigid restraint vibroimpact system with an uncertain parameter. The damping coefficient is considered the uncertain parameter and approximated using arch distribution. The vibroimpact system with the uncertain parameter is firstly transformed into a deterministic system using the Chebyshev polynomial approximation method. The numerical method of the deterministic system is then used to study the bifurcation behavior of the system under asymmetric constraint. The results from the Monte Carlo method and the response of the deterministic system are compared, the period-doubling bifurcation behavior of the system is studied, and the bifurcation behavior of the system is analyzed. The results show that Chebyshev polynomial approximation is an effective method for solving multi-degree-of-freedom systems with uncertain parameters. At the same time, the results of numerical simulations show that the asymmetry of the constraint conditions makes the period-doubling bifurcation of the stochastic constraint system more sensitive to the change in the bifurcation coefficient. The general period response will not change much under the influence of weak random factors. However, at the nodes where period-doubling bifurcation occurs, even small random vibrations can have a significant impact. Furthermore, the method may be further generalized to more general vibroimpact systems with uncertain parameters in future research.

## Figures and Tables

**Figure 1 entropy-25-01365-f001:**
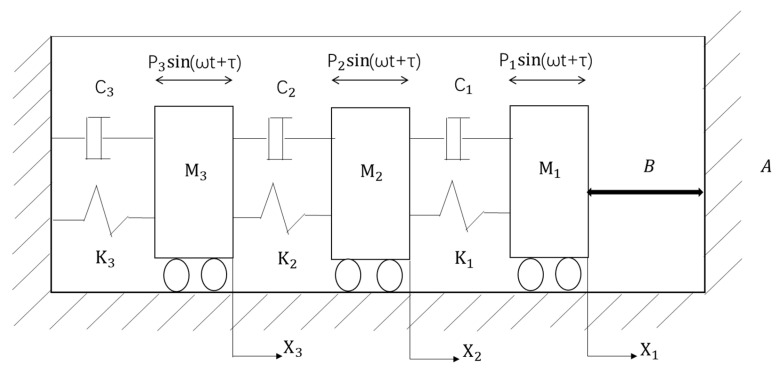
A schematic of the three-degree-of-freedom vibroimpact system with unilateral rigid constrain.

**Figure 2 entropy-25-01365-f002:**
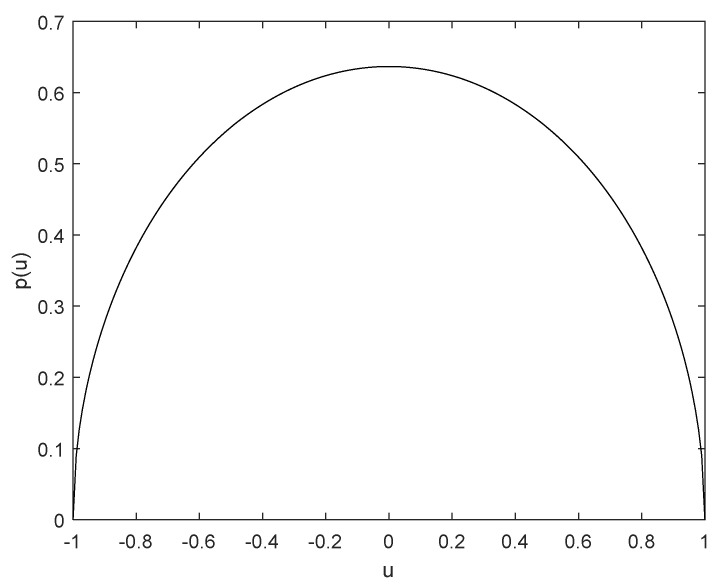
The probability density function curve of arch distribution.

**Figure 3 entropy-25-01365-f003:**
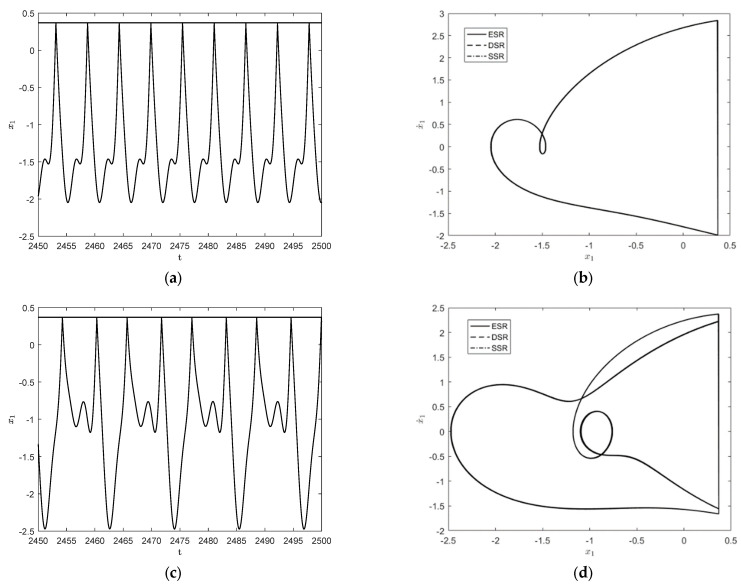
Time history diagrams and phase orbit diagrams of the response for different excitation frequencies: (**a**) time history diagram of deterministic system for ω=2.25; (**b**) the corresponding 1−1 cycles of the three systems for ω=2.25; (**c**) time history diagram of deterministic system for ω=2.2; (**d**) the corresponding 2−2 cycles of the three systems for ω=2.2.

**Figure 4 entropy-25-01365-f004:**
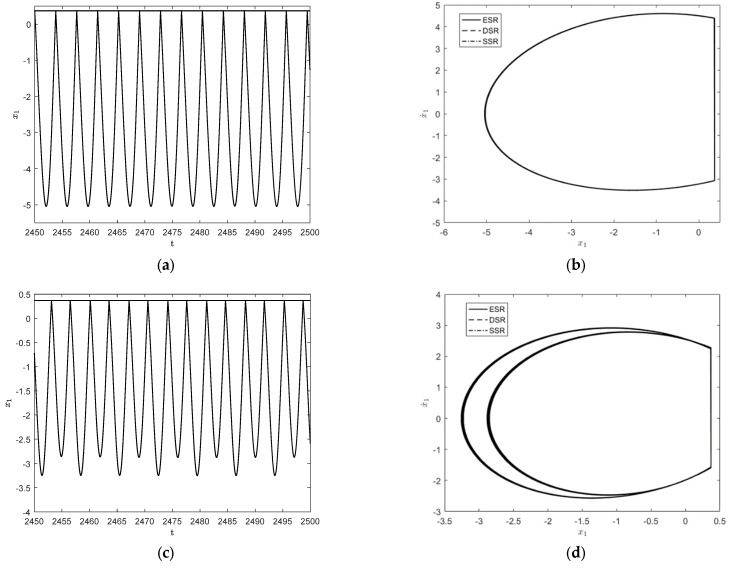
Time history diagrams and phase orbit diagrams of the system responses: (**a**) the time history diagram of the deterministic system for ω=1.65; (**b**) the phase orbit diagram of the three systems in 1−1 cycles for ω=1.65; (**c**) the time history diagram of the deterministic system for ω=1.787; (**d**) the phase orbit diagram of the three systems in 2−2 cycles for ω=1.787.

**Figure 5 entropy-25-01365-f005:**
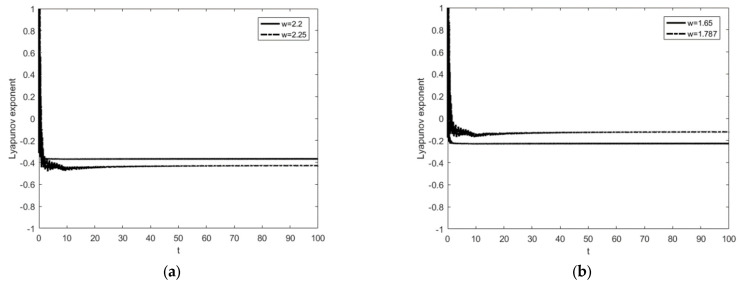
The image of the maximum Lyapunov exponent changing with time. (**a**) ω = 2.2 and 2.25; (**b**) ω = 1.65 and 1.787.

**Figure 6 entropy-25-01365-f006:**
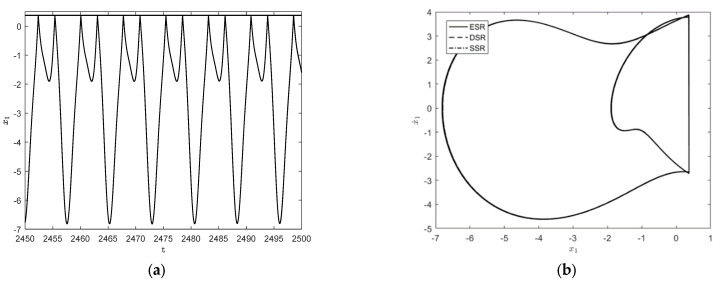
Time history diagrams and phase orbit diagrams of the system responses: (**a**) the time history diagram of the deterministic system for ω=2.45; (**b**) the phase orbit diagram of the three systems in 1−1 cycles for ω=2.45; (**c**) the time history diagram of the deterministic system for ω=2.4; (**d**) the phase orbit diagram of the three systems in 2−2 cycles for ω=2.4.

**Figure 7 entropy-25-01365-f007:**
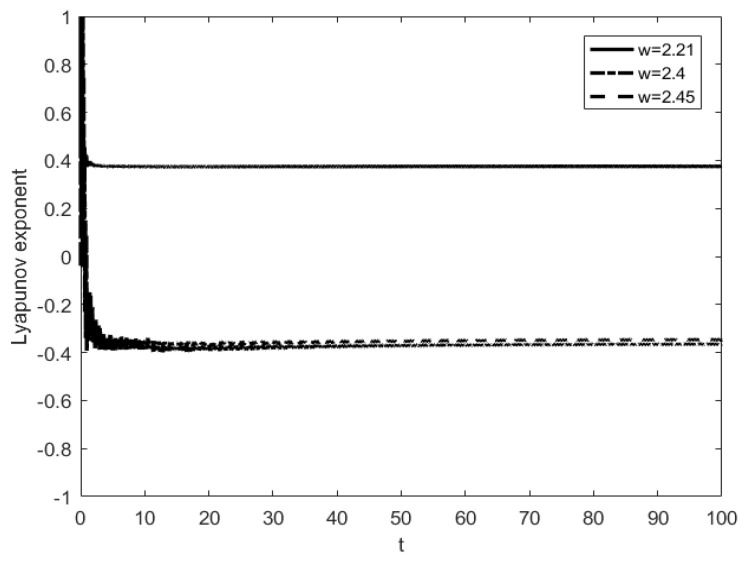
The maximum Lyapunov exponent of period-fold bifurcation to chaos varying with time.

**Figure 8 entropy-25-01365-f008:**
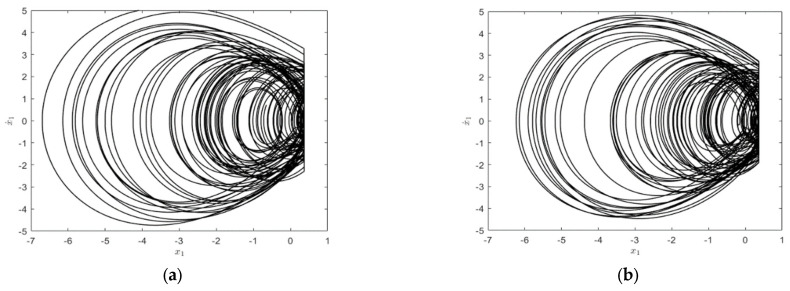
Chaotic phase orbit diagrams and time history diagrams: (**a**) phase orbit diagram of equivalent deterministic system; (**b**) phase orbit diagram of deterministic system; (**c**) phase orbit diagram of original stochastic vibroimpact system; (**d**) time history diagram of deterministic vibroimpact system.

**Figure 9 entropy-25-01365-f009:**
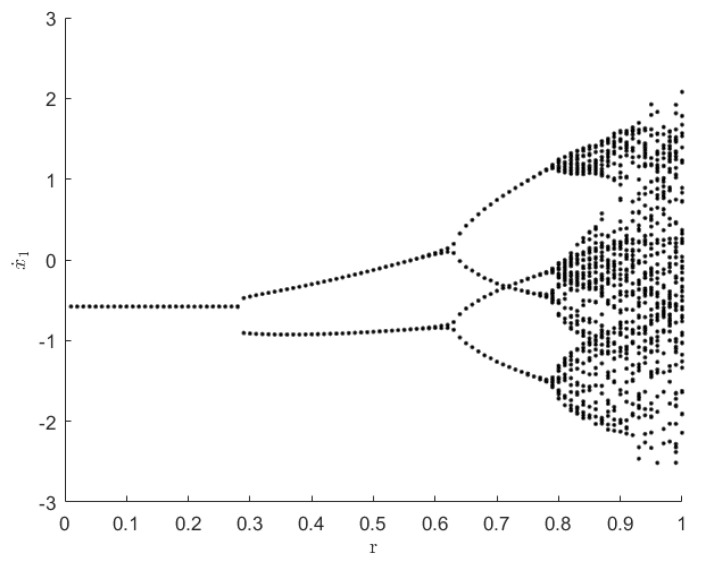
Bifurcation diagram with respect to r for a deterministic three-degree-of-freedom unilateral rigid restraint vibroimpact system.

**Figure 10 entropy-25-01365-f010:**
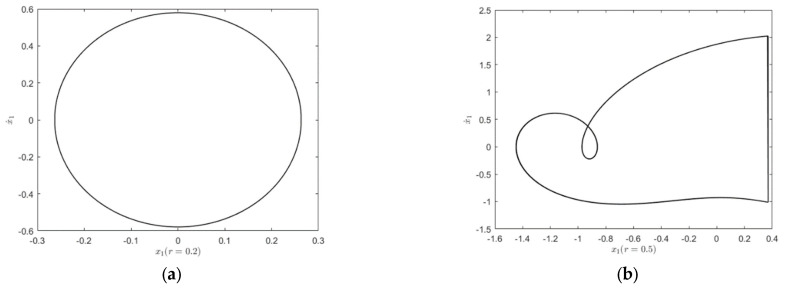
Phase orbital diagram of the system with respect to r: (**a**) r=0.2; (**b**) r=0.5; (**c**) r=0.7; (**d**) r=0.9.

**Figure 11 entropy-25-01365-f011:**
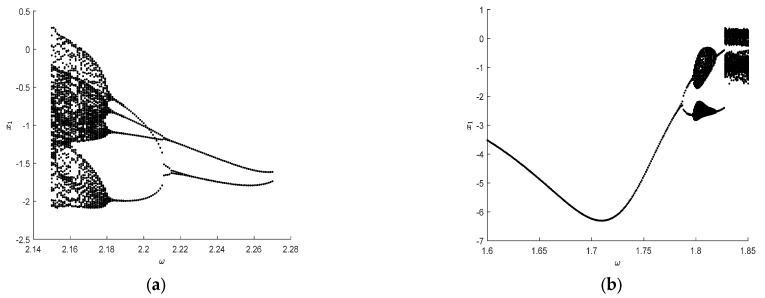
Bifurcation diagram with respect to ω for deterministic three-degree-of-freedom vibroimpact system. (**a**) ω=2.14 to ω=2.28; (**b**) ω=1.6 to ω=1.85.

**Figure 12 entropy-25-01365-f012:**
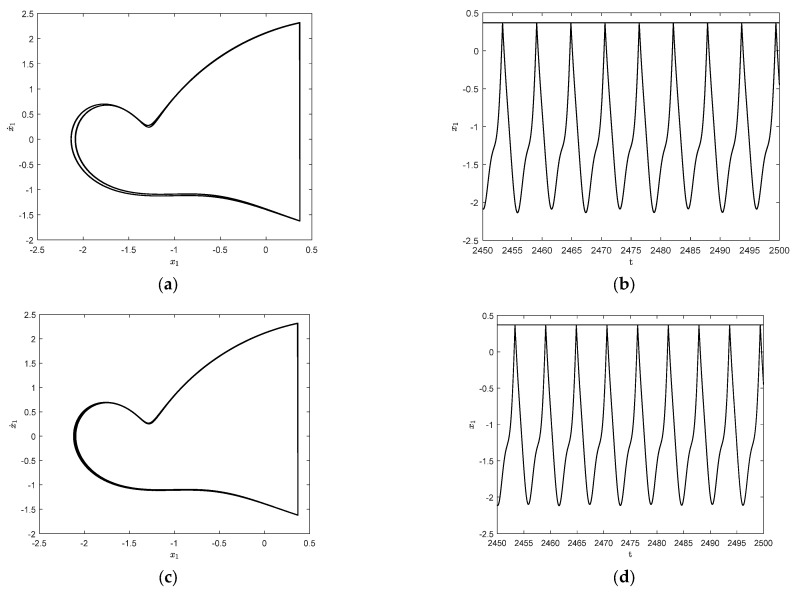
Time history and phase orbit diagrams of stochastic and deterministic systems with ω=2.135: (**a**) phase orbit diagram of original stochastic vibroimpact system; (**b**) time history diagram of original stochastic vibroimpact system; (**c**) phase orbit diagram of the deterministic system; (**d**) time history diagram of the deterministic system.

## Data Availability

The data that support the findings of this study are available from the corresponding author upon reasonable request.

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
