# Peer review of "Response Analysis of the Three-Degree-of-Freedom Vibroimpact System with an Uncertain Parameter"

_entropy, 2023, doi:10.3390/e25091365_

Round 1
Reviewer 1 Report
See attached

Reviewer 2 Report
The present study aims to analyze the dynamical characteristics of the three-degree-of-freedom vibroimpact system with an uncertain parameter by means of the Chebyshev polynomial approximation method. The study has shown some positive results. Some comments:
1) It is a little hard to follow the storyline of the introduction. More remarks should be added to display the difficulties, novelty, and efficiency of the proposed results.
2) Please note that the first time an abbreviation appears, the full name should be given, such as “…simulation using ARM algorithm…”
3) For the results presented in the Figs. 9-12, more explanations on them seem necessary and helpful to readers.
4) How can the research findings guide engineering practice? Please explain.
5) The literature review is not sufficient. Some recently papers about Bifurcation are suggested to be cited. Such as: Hopf bifurcation analysis of maglev vehicle–guideway interaction vibration system and stability control based on fuzzy adaptive theory[J]. Computers in Industry, 2019, 108: 197-209
Reviewer 3 Report
The paper is well written and deserves to be published. The following improvements are recommended to the authors:
1)Point out clearly the novelty of the paper. Are there similar approaches for other models or was the model studied using other approaches?
2) Explain why you chose to study the system of three coupled oscillators. Is it relevant for the real vibro-impact systems?
3) Explain what is the effect on the accuracy of the results by using the cut-off N=2 in the series expansion of the responses.
4) Explain the limits of your results. Does the paper really “establish a model of three-degree-of-freedom unilateral rigid restraint vibroimpact system with an uncertain parameter”, as authors claim? Can the limited numerical simulations done by authors lead to general conclusions on the dynamical behavior of the system?
5) There are unclear formulations that should be revised, as for example:
- At the end of the first paragraph from 4.1: “For vibroimpact systems, the Jacobian matrix discontinuity caused by impact factors exists in the model, the calculation method of smooth system cannot be directly applied to the system.” could be reformulated as “For vibroimpact systems, due to the discontinuity induced in the Jacobian matrix by impact factors, the calculation method of smooth system cannot be directly applied to the system.”
- In the next paragraph, when a set of values for the parameters are taken, both the restitution coefficient and impact boundary are considered as r=0.7. 6) The graphical representations on the same diagrams for the three systems - DSR, SSR and ESR, should be improved. The curves cannot be differentiated.
Small reformulations are required
